# Unraveling the Bioactive Potential of *Camellia japonica* Edible Flowers: Profiling Antioxidant Substances and In Vitro Bioactivity Assessment

**DOI:** 10.3390/ph17070946

**Published:** 2024-07-15

**Authors:** Antia G. Pereira, Maria Fraga-Corral, Aurora Silva, Maria Fatima Barroso, Clara Grosso, Maria Carpena, Pascual Garcia-Perez, Rosa Perez-Gregorio, Lucia Cassani, Jesus Simal-Gandara, Miguel A. Prieto

**Affiliations:** 1Nutrition and Bromatology Group, Department of Analytical Chemistry and Food Science, Instituto de Agroecoloxía e Alimentación (IAA)—CITEXVI, Universidade de Vigo, 36310 Vigo, Spain; mfraga@uvigo.es (M.F.-C.); mass@isep.ipp.pt (A.S.); mcarpena@uvigo.es (M.C.); pasgarcia@uvigo.es (P.G.-P.); mariarosa.perez@uvigo.es (R.P.-G.); luciavictoria.cassani@uvigo.es (L.C.); jsimal@uvigo.es (J.S.-G.); 2REQUIMTE/LAQV, Instituto Superior de Engenharia do Porto, Instituto Politécnico do Porto, Rua Dr António Bernardino de Almeida 431, 4249-015 Porto, Portugal; mfb@isep.ipp.pt (M.F.B.); claragrosso@graq.isep.ipp.pt (C.G.); 3Department for Sustainable Food Process, Università Cattolica del Sacro Cuore, Via Emilia Parmense 84, 29122 Piacenza, Italy; 4LAQV-REQUIMTE Department of Chemistry and Biochemistry, Faculty of Sciences, University of Porto, Rua do Campo Alegre s/n, 4169-007 Porto, Portugal

**Keywords:** *Camellia japonica*, petals, extraction, characterization, valorization, phytochemicals

## Abstract

In recent years, the search for novel natural-based ingredients by food and related industries has sparked extensive research aimed at discovering new sources of functional molecules. *Camellia japonica*, traditionally known as an ornamental plant, has gained attention due to its diverse array of bioactive compounds with potential industrial applications. Although *C. japonica* flowers are edible, their phytochemical profile has not been thoroughly investigated. In this study, a phenolic profile screening through an HPLC–ESI-QQQ-MS/MS approach was applied to *C. japonica* flower extracts, revealing a total of 36 compounds, including anthocyanins, curcuminoids, dihydrochalcones, dihydroflavonols, flavonols, flavones, hydroxybenzoic acids, hydroxycinnamic acids, isoflavonoids, stilbenes, and tyrosols. Following extract profiling, their bioactivity was assessed by means of in vitro antioxidant, antimicrobial, cytotoxic, and neuroprotective activities. The results showed a multifaceted high correlation of phenolic compounds with all the tested bioactivities according to Pearson’s correlation analysis, unraveling the potential of *C. japonica* flowers as promising sources of nutraceuticals. Overall, these findings provide insight into the valorization of *C. japonica* flowers from different unexplored cultivars thus diversifying their industrial outcome.

## 1. Introduction

The growing trend in consumer preference for natural-based products, avoiding synthetic additives, has led to a paradigm shift in various industries, including food, pharmaceuticals, and cosmetics. This transformative trend underscores the imperative to explore innovative, natural, and sustainable sources of health-promoting ingredients [1].

Specifically, compounds with antioxidant properties are currently in high demand, sought not only for their efficacy in extending product shelf life but also for their pivotal role in neutralizing free radicals. This functionality is crucial in preventing a spectrum of disorders and diseases, including cancer, cardiovascular diseases, neurodegeneration, hypertension, and diabetes mellitus [2]. Phenolic compounds, renowned for their antioxidant capabilities, frequently manifest additional bioactivities, encompassing anti-inflammatory, antimicrobial, and anti-proliferative effects [2]. This diverse spectrum of bioactivities positions phenolic compounds as pivotal molecules with a myriad of applications across diverse industries. Consequently, there has been a surge in investigations over the last decade focusing on evaluating new natural sources of phenolic compounds.

In this context, *Camellia* genus (family Theaceae) has emerged as a potential alternative. Comprising approximately 280 species native to Asia, this globally distributed genus is considered economically significant. Among their most popular species, *Camellia sinensis* (tea beverages) and *Camellia japonica* (ornamental), also called winter rose, stand out. Particularly, over 10,000 cultivars and 3000 hybrids of *C. japonica* are currently known, with a cultivar of beautiful flowers of different colors and shapes [3,4]. The ornamental camellias were brought to Europe and Americas in late 1870s and are now popular flowering and landscaping shrubs in many regions across the world with a mild climate, such as Northwestern Spain [4,5]. Therefore, nowadays, they are globally distributed and considered an economically important group of perennial evergreen flowering plants [6].

Regarding the chemical composition of *Camellia* sp., the most abundant biomolecules are phenolic compounds, terpenoids, fatty acids, and a few key minor compounds that include pigments and biosugars [7]. This rich chemical profile, mainly associated with *Camellia* leaves, has prompted its traditional use as medicine in Asia (Japan, China, Korea) for the treatment of stomachic illness, bleeding, and inflammation [8]. Nowadays, this traditional application as therapy has been supported by several studies that have demonstrated the beneficial properties of *Camellia* sp., such as antioxidant, anti-inflammatory, or antimicrobial properties, among others [7,9].

Therefore, this species may represent not only a source of antioxidant phenolic compounds, but also a source of compounds with antimicrobial activity. This is of great interest since the antibacterial potential of plant-derived molecules has gained particular relevance in addressing the escalating challenge of antibiotic resistance—a significant concern in public health [10]. Antibiotic resistance not only poses a substantial threat to public health but has also been suggested to impede advancements in cancer treatment, contributing to the escalating incidence of cancer cases worldwide [11,12]. Within this context, the antiproliferative capacity of bioactive compounds has emerged as a promising attribute, offering alternative avenues for treatment [13].

Despite these advancements, the flowers of *C. japonica*, the most representative species in the study area, lack studies analyzing the use of this raw material. While information on the bright and large flowers of this genus is scarce compared to leaves and seeds, they present an attractive option for research, given their ability to synthesize phenolic compounds and other bioactive secondary metabolites [14]. *Camellia* flowers are considered to possess astringent, antihemorrhagic, hemostatic, and salve properties [15]. Despite these health-enhancing bioactivities, camellias are still considered an underexploited raw material, especially regarding their flowers. Therefore, greater efforts are needed to fully achieve their chemical and bioactive characterization.

In this work, extracts of eight cultivar of *C. japonica* flowers obtained by an easy and profitable extraction technique (heat assisted extraction) have been analyzed to determine their chemical profile and assess their bioactivities with the aim of providing the basis for their further exploitation as sustainable sources natural ingredients.

## 2. Results and Discussion

### 2.1. Phytochemical Profile

Flower extracts from the different *C. japonica* cultivars underwent phenolic profiling using HPLC–MS/MS. This technology enabled the identification of a total of 36 distinct phenolic compounds. Table 1 shows the mass data and the tentative identity of the peaks. A total of 36 peaks were cautiously identified, corresponding to anthocyanins (10), curcuminoids (1), dihydrochalcones (3), dihydroflavonols (4), flavonols (2), flavone (3), hydroxybenzoic acids (2), hydroxycinnamic acids (3), isoflavonoids (5), stilbenes (1), and tyrosols (2). Hydroxybenzoic acids (7.92–44.23% of mass of total compounds identified), namely ellagic acid acetyl-arabinose and gallic acid 4-*O*-glucoside, and tyrosols (23.11–55.38%), oleuropein and demethyloleuropein, were the main compounds identified.

The high presence of hydroxybenzoic acids contrasts with the previous results from experiments performed with *C. japonica* flower extracts [16,17], which supports the idea that *C. japonica* can be a promising source of this type of compound as reported by Pereira et al. (2023) [18]. Regarding the class of tyrosols, the most abundant molecule found in this study was oleuropein, with concentrations ranging from 10 to 79 μg per 100 g of extract. In prior investigations, the concentration of tyrosols accounted for 1.1% of the peak area in *C. japonica* petal wine, designating them as samples rich in potential antioxidative volatiles [19]. Additionally, it was noted that such tyrosol-rich extracts could potentially exhibit utility in mitigating diabetes [19]. Furthermore, it has been demonstrated that tyrosol-rich extracts obtained from camellias exhibited a superior antioxidant effect in camellia oils compared to exogenous reference antioxidants such as caffeic acid [20]. Therefore, the presence of these compounds in camellia extracts is of great interest, as they not only hold the potential for bioactivities but also contribute to enhanced preservation of the extracts [20]. Nevertheless, a scarcity of studies persists in documenting the presence of tyrosol in *C. japonica* raw materials.

Anthocyanins, which have been previously reported in *C. japonica* flowers, are responsible for the different colorations of camellia cultivars, as supported by different studies [18,21]. Nevertheless, the molecular mechanisms governing flower pigmentation in camellias remain unexplored, and the current breeding program for camellias relies on serendipitous seedling selection. Molecular insights into the regulation of camellia flower traits are essential for the development of an effective and targeted breeding system for the plant [22]. This lack of understanding, coupled with the extensive diversity of *C. japonica* cultivars and their diverse colorations, leads to widely varying results in the available studies on anthocyanin content, both in terms of reported molecules and their concentrations. In this regard, previous studies have identified cyanidin 3-*O*-glycosides [23], cyanidin 3–galactoside [22,24], cyanidin 3-*p*-coumaroylglucoside [25] and leucoanthocyanins [26,26] as the primary anthocyanins in *C. japonica*. In other species of the same genus, such as *C. sinensis*, cyanidin, malvidin, petunidin, and pelargonidin have also been reported as the main anthocyanins among cultivars with pink flowers [27]. These findings align with the results of our study (Table 1). It was observed that Eugenia de Montijo (EM) exhibited the most vibrant pink coloration according to the color chart of the Royal Horticultural Society of London (RHS). This vibrant pink coloration could be attributed to its significantly higher content of anthocyanins, including peonidin, peonidin derivative, malvidin 3-*O*-(6″-caffeoyl-glucoside), and cyanidin 3-*O*-glucosyl-rutinoside, in comparison to the other cultivars studied. Nevertheless, our study also noted a limited presence of pelargonidin glucosides in *C. japonica*, consistent with the findings of previous authors [23]. In all instances, the camellia samples consistently exhibited a noteworthy presence of anthocyanins, comprising a substantial proportion of the total identified compounds (ranging from 4.16% to 17.13% of the mass of the total compounds). This class of bioactive compounds has been recognized for its significant biological properties, including potent antioxidant, anti-tumor, and neuroprotective effects [28]. Therefore, this chemical composition could partly justify the biological effects reported in this raw material.

In addition to anthocyanins, the flavonoid profile of *C. japonica* flowers includes the presence of three flavones, including two apigenin glycosides and nobiletin (Table 1). Each sample manifested a distinctive flavone profile, underscoring the chemical diversity among various cultivars. Notably, CT and HA demonstrated elevated concentrations of nobiletin, positioning them as significant contributors of this flavone. EM stood out for its high content in apigenin 7-*O*-glucuronide. While prior studies conducted on flowers and fruit shells of other *C. japonica* cultivars indicated only minimal concentrations of flavones [23,29], more recent research involving the cultivars examined in this study reveals substantial levels [18]. Among flavonols, the two compounds identified were quercetin-3-*O*-arabinose and kaempferol 3-*O*-acetyl-glucoside, both compounds having been previously reported in *C. japonica* var Eugenia de Montijo flowers [30]. The presence of these compounds was previously reported in the leaves of *C. japonica* [8], with concentrations ranging from 10.4 to 75.1 mg catechin/g dry leaf, depending on the cultivar studied. In these leaves, the predominant compounds are three aglycone flavonoids (quercetin, kaempferol, and apigenin) and two glycosylated flavonoids (rutin and quercitrin) [31]. However, it is anticipated that the concentration of flavonoids in *C. japonica* leaves is higher than in the flowers due to the extensive body of research emphasizing tea leaves (produced from various *Camellia* sp.) as a rich source of flavonoids [32,33].

Regarding dihydrochalcones, this is a compound family previously documented in camellias [18,34]. In the case of these camellia cultivars, dihydrochalcones do not represent a predominant group of compounds, comprising only 0.15–0.80% of the total identified compounds. This observation is consistent with findings from previous studies involving the *Camellia* species, where dihydrochalcones were also not identified as major compounds [18,35]. However, it is noteworthy that one of the identified compounds, phloridzin, which had previously been reported in *C. japonica* [36,37], holds significance for the commercialization of camellia products. This is because phloridzin is recognized as one of the primary sweet taste compounds, and its concentration plays a vital role in ensuring tea quality. The recognition threshold for phloridzin is 0.01 mg/mL, and it demonstrates synergistic effects when combined with dulcitol, l-alanine, and sucrose [38,39].

Within the dihydroflavonol family, four isomers of dihydroquercetin were successfully isolated. These compounds had previously been identified in extracts from *C. japonica* var. Eugenia de Montijo (EM) using various extraction techniques, including ultrasounds and microwave [30]. Previous research in different *Camellia* species, such as *Camellia nitidissima* and *C. japonica*, regarded this compound as an intermediate in the production of color-inducing compounds like quercetin, cyanidin, and their respective derivatives [40,41]. This compound was also associated with high antioxidant, capillary-protective, and anti-inflammatory activity [42]. Therefore, it could be interesting to conduct an optimization of the extraction to increase phenolic compound yield.

Regarding the disparities observed among the different cultivars investigated in this study, the EM cultivar emerged as particularly noteworthy due to its elevated concentrations of anthocyanins, dihydrochalcones, hydroxybenzoic acids, curcuminoids, hydroxycinnamic acids, and stilbenes. Notably, the content of stilbenes was at least six times higher, while dihydrochalcones and hydroxybenzoic acids were a minimum of three times more abundant in this cultivar compared to others. As a result, this cultivar is a promising candidate for future optimization studies. The CT and HA cultivars are notable for their substantial dihydroflavonol content, which was ten times higher than that of the other cultivars considered in this study. The EM and CT cultivars are the richest in flavonols and flavones, with a content at least three times higher for flavanols. In terms of isoflavonoids, the cultivars with the highest concentration are EM, CT, and HA, boasting a content at least two times greater than that of the other cultivars. Additionally, the CT cultivar stands out for its high tyrosol content, with concentrations reaching up to eight times those found in the other cultivars. These findings underscore the potential of EM, CT, and HA as valuable candidates for future optimization studies.

In addition, for future investigations, it would be valuable to assess the presence of bound phenolic compounds in the samples, also referred to as non-extractable, unextractable, or insoluble phenolics [43]. Generally, these compounds are ester-linked to structural cell wall polymers. Liberation of these compounds typically occurs through alkali, acid, or enzymatic treatment of samples prior to extraction [44]. However, operations conducted with the extracts, such as extrusion for incorporating the extracts into a food matrix, could also result in the release of these bound phenolics [45]. This is of considerable interest, as various studies substantiate that the primary contributor to antioxidant activity in both raw and processed vegetables is in the bound form. Thus, the potential of these plants as antioxidant agents may be even greater, considering that these compounds were not considered in this study (due to the removal of such large compounds by the used filters). Nonetheless, it is important to note that the bioaccessibility of bound phenolics is lower than that of free compounds [45,46]. Therefore, understanding the precise composition of bound phenolics is crucial for evaluating their physicochemical properties, nutritional values, potential applications, and for conducting epidemiological and clinical studies addressing their potential health effects.

### 2.2. Antioxidant Capacity

The antioxidant capacity of the different cultivars of *C. japonica* analyzed shows a significant variability between them (Table 2). The DPPH values of the cultivars under study oscillate between 29.71 and 39.18 µg/mL for Eugenia de Montijo (EM) and Conde de la Torre (CT), respectively. These results are significantly lower than those reported in previous studies involving these specific cultivars, in which values as high as 136.5 µg/mL were reported for the Elegans variegated cultivar [18]. Therefore, all samples can be considered potential antioxidant agents. These values are similar to those reported in other studies, with DPPH values of 45 µg/mL in *C. japonica* flowers [47] and 38.53 µg/mL in *C. japonica* leaves [8]. However, our results are lower than the outcomes obtained from the pink petals of *C. japonica* extracted with 0.1% of trifluoroacetic acid in ethanol (3.8 μg/mL), while the red and white petals showed higher values (7.31 and 43 μg/mL, respectively) [48]. Similar outcomes were obtained for the purified red pigments extracted from *C. japonica* flowers that showed DPPH scavenging activity comparable to that of the standard of butylated hydroxyanisole (BHA) (4.55 and 4.17 μg/mL, respectively) [49]. These values, significantly better than those obtained in this study, may be due to the utilization of the purified pigments in Zhang’s study. However, other purified fractions of *C. japonica* such as polysaccharides still showed similar or even higher values (36–119 µg/mL) than those from the present work [50]. Regarding other species of the same genus, such as black tea, *C. sinensis*, the antioxidant results are two orders of magnitude higher even when different solvents such as the following are used: 9 mg/mL for 50% MeOH or 42 mg/mL for water [51]. While an aqueous extract obtained from *C. nitidissima* leaves and further fractionated using several solvents showed values similar to ours (from 37 to >60 µg/mL) [52]. In general terms, the antioxidant potential of *C. japonica* is higher than other plants recognized as medicinal like *Terminalia glaucescens* (145.54 μg/mL) [53], *Zygophyllum album* (96 μg/mL), [54], or *Sida rhombifolia* (546.1–1222.5 μg/mL) [55].

The ABTS values of the cultivars analyzed ranged from 77.23 to 120.82 μg/mL (CT, Conde de la Torre and TU, Carolyn Tuttle, respectively). According to the previous reports, the *C. japonica* flower ethanolic extracts had a better ABTS radical scavenging capacity than vitamin C at the same concentration (50 μg/mL) [47]. These values were significantly better than those obtained with other parts of *C. japonica*, such as the roots, which showed ABTS values in the range of 200–2000 µg/mL [56] and the leaves, with an ABTS scavenging activity of 96.10 mg/mL [57]. Other species such as *C. nitidissima* showed values ranging from 15 to 24 μg/mL [52]. However, the high variability observed from the experimental conditions of ABTS assay, together with the variability regarding the presentation of the results, hinders the proper comparison of the results. For instance, several works have provided data in different units, such as in the case of Zheng’s study where flowers of *C. japonica* were proven to have an ABTS activity of 165.89 μmol Trolox equivalents/g sample [58], or in the case of Luo’s study where different species of the genus *Camellia* were assessed but the results were provided as percentages (7.79–89.35%) [59].

### 2.3. Antioxidant Activity

Regarding the comparison of CBA, SRSA, OHSA, NOSA, and H_2_O_2_, results are scarcer since these assays seem not to have been widely applied for the study of *Camellia* sp. Concerning the crocin bleaching inhibition assay (Figure 1), CBA values of the analyzed cultivars ranged from 0.01 to 0.43 min/g dw for Grandiflora Superba (GS) and Hagoromo (HA), respectively. The lower activity values were obtained from the Grandiflora Superba (GS) and Carolyn Tuttle (TU) (Table 2). As this is not a widely used method, its comparison with the results obtained from other camellias or plant matrices has not been possible. The SRSA values of the different cultivars analyzed in this study ranged between 0.02 and 0.11 mg/mL (Figure 2), with significant differences among the samples. EV, DD, AA, and TU showed no significant differences, with their EC_50_ values near 0.02, accounting for the best results for SRSA. These outcomes are much lower than the ones reported in previous studies, such as in the case of the fermented *C. japonica* leaves (IC_50_ 0.23 mg/mL) [60]. However, purified polysaccharides from the same species provided values within our range (0.043 to 0.067 mg/mL) [50], while the SRSA values for the petals extracted with an acidified ethanol solution were much lower at around 0.003 mg/mL [48]. The OHSA values obtained in our work (EC_50_ from 0.6 to 1.3 mg/mL) are slightly lower than those reported in previous studies for polysaccharides from *C. oleifera*, which were in the range of 1.3–3.7 mg/mL [61], or those from *C. japonica* that ranged from 0.32 to 1.04 mg/mL [50]. Only one study showed much lower OHSA results from analyzing colorful petals extracted under acidified conditions, with values ranging between 6.2 and 12.6 μg/mL [48]. DT and CT showed the lowest values and had no significant difference among them. The NOSA data obtained in the present work (EC_50_ from 0.6 to 2.2 mg/mL) are higher than those reported in previous studies with *C. japonica* stems, in which a NOSA value of 0.35 mg/mL was detected [60]. H_2_O_2_ scavenging activity ranged from 0.031 to 0.161 mg/mL for Hagoromo (HA) and Carolyn Tuttle (TU), respectively. These values are in the same range than those obtained with *C. sinensis* flowers, which had IC_50_ values of 6.09–169 μg/mL depending on the solvent used during the extraction [62]. In general terms, our results, which are mainly in concordance with the other previously published results, point to the different cultivars of *C. japonica* flowers as potential sources of antioxidant agents.

### 2.4. Antimicrobial Activity

Among the eight cultivars of *C. japonica,* Dr Tinsley (DT), Eugenia de Montijo (EM), and Conde de la Torre (CT) were demonstrated to be the most effective ones for the inhibition of bacterial growth (Table 2). These three cultivars could inhibit all bacterial species except for *E. coli.* However, other studies carried out with *C. japonica* flowers have proven that some cultivars of *C. japonica* have antimicrobial activity against *E. coli* with inhibition halos of 17–18 mm [63]. Similar values were reported for *C. sinensis* (16.66–23.33 mm) [64]. Less common camellia species, such as *C. assamica*, have shown lower levels of activity against *E. coli* (3–4.3 mm) [65].

The strongest inhibiting zones were observed against *S. epidermidis*, with diameters ranging between 10.9 and 14.0 mm. To the best of our knowledge, these are the first results reported of *C. japonica* activity against *S. epidermidis.* The only other record found in the scientific literature was with regard to magnesium oxide nanoparticles (MgO-NPs) synthesized with the leaf extract of *C. sinensis* (5–25 μg/mL), which exhibited antibacterial activity (10–19 mm inhibition zone) against *S. epidermidis* [66].

On the contrary, the smallest halos were achieved for *B. cereus* (5.1–7.4 mm) although this strain was attacked by two additional cultivars, Donation dentada (DD) and Hagoromo (HA), which showed a broader halo of ~9 mm (Table 2). These inhibition halos were lower than those obtained with other camellia species such as *C. sinensis,* with an inhibition zone of 25.38 mm even when using a lower concentration extract (5 mg/mL) [67]. However, this study used an organic solvent, methanol, for extraction which may have recovered biomolecules with different biological properties. In addition, the concentration of biocompounds may vary notably among tea species; indeed, *C. sinensis* has been suggested to be more effective against Gram-positive bacteria than *C. japonica* [68]. The correlation analysis allowed us to determine a correlation of 0.88 between the activity against this microorganism and the CBA assay (Figure 3).

The only bacterial strain inhibited by all the *C. japonica* cultivars was *P. aeruginosa*, with inhibiting halos of around 9–10 mm, except for the Elegans variegated cultivar (EV), whose halo was around 7 mm. Indeed, this cultivar was just capable of inhibiting the growth of *P. aeruginosa.* In fact, this cultivar, Elegans variegated, displayed no inhibition against the remaining bacterial strains. Within the previously published studies, some cultivars of *C. japonica* had already shown their effectiveness against *P. aeruginosa.* A greater activity was described with halos from 13 to 25 mm, which were dependent on the solvent used, with acetone displaying the strongest inhibition [64].

Among all the analyzed cultivars, only two have been proven to be ineffective against *S. aureus*, namely Elegans variegated (EV) and Donation dentada (DD). The rest of the cultivars had inhibition zones in the range of 9.37–10.98 mm, values slightly lower than those reported by Kim (14–15 mm) [63]. This may have been due to the higher percentage of methanol used in Kim’s study, since other studies have shown that ethanol extracts tend to exhibit higher antimicrobial activity [60]. In another study, the minimum inhibitory concentration (MIC) against *S. aureus* of young leaf *C. japonica* extract were revealed to be around 15 μg/mL [69], while for *C. sinensis*, the MIC was much lower at 0.23 mg/mL [64], which again supports the stronger capacity of *C. sinensis* against Gram-positive bacteria.

### 2.5. Cytotoxic Activity

The different extracts of *C. japonica* flowers showed significant inhibitory effects on the proliferation of different cancerous cell lines (AGS, HepG2, A549) (Table 2). The highest cytotoxicity was observed against A549, whose IC_50_ never exceeded 22.2 µg/mL, whereas the IC_50_ values were 8.69–51.67 µg/mL for the AGS cells and 13.37–45.05 µg/mL for the HepG2 cells. Data obtained in the previous studies reported similar levels of activity of the *C. japonica* flower extracts against the AGS cells, with IC_50_ values of 14.1–79.4 µg/mL, depending on the cultivar analyzed [18]. In contrast, all the cultivars had cytotoxic effects on non-tumor Vero cells, with IC_50_ values ranging between 6.23 and 18.40 µg/mL. CT (Conde de la Torre) and GS (Grandiflora Superba) showed the lowest toxicity against Vero cells (IC_50_ = 51.67 and 68.99 µg/mL, respectively). From the results, six cultivars, Elegans variegated (EV), Dr Tinsley (DT), Eugenia de Montijo (EM), Donation dentada (DD), Hagoromo (HA), and Carolyn Tuttle (TU), showed significantly more cytotoxic activity against the cancer cell lines at different degrees. Cultivar DT (Dr. Tinsley) showed the lowest IC_50_ = 1.65 µg/mL and thus had the best cytotoxic activity against A549 cells, improving the capacity of the ellipticine used as positive control. This high activity may have been due to the presence of catechins, a compound present in the *Camellia* genus that has been shown to have cytotoxicity against A549 cell line, even though units of data hinder their comparison (inhibitory rate of 19.76% at a concentration of 600 µmol/L, 24 h incubation) [70]. This activity may also have been due to the high concentration of saponins of the *Camellia* genus [71]. Data obtained from other species of camellias revealed IC_50_ values of 64.4–113.3 mg GAE/L for *C. sinensis* [72], and from <10 µM [73] to 18.5 μM [74] for *C. oleifera*, although the variability of units once again do not allow the comparison of the cytotoxic effect.

To our knowledge, there are no studies that analyze the cytotoxic activity of *C. japonica* against the other cell lines used in this study. However, *C. oleifera* [75] and *C. ptilophylla* (IC_50_ 292 μg/mL at 72 h) [76] have been proven to be effective against the HepG2 cell line. *C. sinensis* has no significant activity against HepG2 in comparison with the standard (doxorubicin, IC_50_ of 18.8 µM) [77]. Therefore, *C. japonica* seems to have more potential as a source of cytotoxic compounds than the other *Camellia* species, especially the cultivars DT (Dr Tinsley) and DD (Donation dentada) against the cell line A549. Nevertheless, *C. japonica* has less activity than the other reference compounds used against the same cell line.

However, it is imperative to persist in this line of research, as recent studies have demonstrated how phenolic compounds in food undergo a series of alterations by the intestinal microflora and enterocytes. This process leads to the presence of molecules in the plasma that possess distinct molecular structures and bioactivity compared to the original molecules present in the consumed foods. This concept holds significant relevance, especially in the context of studies involving cultured cells, particularly liver and lung lines, in which the phenolic compounds were supplemented that are not actually present at the level of these tissues in vivo [78].

### 2.6. Neuroprotective Effects

The data obtained in this study show low neuroprotective activities for the analyzed samples, making it impossible to determine EC_50_ (Table 2). However, previous studies have reported that some noroleanane triterpenoids isolated from the fruit peels of *C. japonica* had potential neuroprotective and anti-inflammatory effects against Aβ-induced neuronal damage [79]. Other cultivars of camellias (*C. nitidissima*) have also been proven to have a neuroprotective effect via synergistically boosting the endogenous antioxidant defenses and neurotrophic signaling pathway. The main active phytochemicals seemed to be catechins, quercetin, kaempferol, and their derivatives [80]. Green tea from *C. sinensis* presented a strong neuroprotective capacity in the Alzheimer-like rat model, avoiding memory deficits and increases in ROS and TBARS levels in the hippocampus. This activity was attributed to the high content of epigallocatechin gallate [81]. Therefore, in our case, it could be interesting to purify/isolate the compounds present in the sample to better assess this bioactivity.

### 2.7. Pearson’s Correlation Analysis

Additionally, Pearson’s correlation analysis was performed to elucidate the concealed patterns and relations between the different bioactivities assessed (Figure 3). According to these results, antimicrobial activity may be due to the presence of high concentrations of antioxidant compounds. This can also be observed in Table 2, in which the camellias with the highest antioxidant capacity are EM (Eugenia de Montijo) and HA (Hagoromo), which are also the ones with the highest inhibition halos. This correlation between antioxidant and antimicrobial activity had already been observed in previous studies with other vegetal matrices, such as *Myrtus* spp., *Eucalyptus oleosa*, and rosemary [82,83].

## 3. Materials and Methods

### 3.1. Chemicals and Reagents

Ethanol was purchased from VWR (Radnor, PA, USA). All organic solvents utilized for extraction and chromatographic analysis were of HPLC-grade quality. To ensure high purity of the water used, water was obtained from a Direct-Q 5UV Millipore equipment (Merck, Rahway, NJ, USA). Folin–Ciocalteu and 2,2-diphenyl-1-picryl-hydrazyl-hydrate free radical scavenging (DPPH) were purchased form Cientisol (Santiago de Compostela, Spain). Dimethyl sulfoxide (DMSO), lactic acid, ascorbic acid, 2′,2′-azobis (2-amidinopropane) dihydrochloride (AAPH), butyrylcholinesterase (BuChE), acetylcholinesterase (AChE), 4-nitrophenyl-α-D-glucopyranoside, Tris-HCl, potassium phosphate monobasic, potassium phosphate dibasic trihydrate, bovine serum albumin (BSA), and galantamine were purchased from Sigma-Aldrich (Steinheim, Germany). Sodium nitroprusside was from Labkem (Barcelona, Spain). Sodium carbonate was from Panreac (Barcelona, Spain). Potassium phosphate buffer was from Scharlab (Barcelona, Spain). The culture media, Mueller–Hinton Broth (MHB) and Mueller–Hinton Agar II, were acquired from Sigma-Aldrich (Madrid, Spain). The culture media, Mueller–Hinton Agar II, was acquired from Biolife Milan, Italy. *Staphylococcus aureus* (ATCC 25923), *Bacillus cereus* (ATCC 14579), *Pseudomonas aeruginosa* (ATCC 10145); and *Salmonella enteritidis* (ATCC 13076) were provided by Selectrol, Buckingham, UK; *Escherichia coli* (NCTC 9001), and *Staphylococcus epidermidis* (NCTC11047) were from Microbiologics, Minnesota USA. The cell lines, including the vero cell line, human gastric cancer cell line (AGS), human hepatocellular carcinoma (HepG2), and human lung cancer (A549), were provided by Frilabo (Porto, Portugal). The standards for phenolic compounds, namely cyanidin, luteolin, quercetin, gallic acid, p-coumaric acid, and resveratrol, were procured from Sigma (Saint Louis, MO, USA). Nylon syringe filters with a diameter of 25 mm and a pore size of 0.22 μm were provided by Filter-Lab (Barcelona, Spain). All the spectrophotometric assays were performed in a Synergy HT W/TRF Multi Mode Microplate Reader with Gen5 2.0software (BioTek Instruments, Winooski, VT, USA).

### 3.2. Camellia Sampling and Preparation

Studies were conducted using eight different cultivars of 20-year-old *C. japonica* flowers, four of them had the white phenotype (Conde de la Torre, Dr Tinsley, Grandiflora Superba, Hagoromo) and four had the pink phenotype (Elegans variegated, Donation dentada, Eugenia de Montijo and Carolyn Tuttle). All of them were manually harvested in NW Spain (42.431° N, 8.6444° W) in January 2020 at their optimal state of maturity (stage 5, full bloom), harvesting a total quantity of 400 g of fresh flowers (around 15 flowers for each cultivar). The different flowers were botanically identified by official germplasm banks and identification guide resources, including the American Camellia Society, the International Camellia Register, and previous studies of identification [84]. Fresh flowers were immediately frozen at −80 °C to prevent their degradation. Once frozen, they were lyophilized (LyoAlfa 10/15 from Telstar, Thermo Fisher Scientific, Shanghai, China) and pulverized into a fine powder (~63 μm). The resulting powder was mixed to guarantee the homogeneity of the samples and stored at −20 °C until their analysis. To facilitate experimental work and data analysis, each cultivar was named as follows: *EV*, Elegans variegated; *DT*, Dr Tinsley; *EM*, Eugenia de Montijo; *CT*, Conde de la Torre; *DD*, Donation dentada; *HA*, Hagoromo; *GS*, Grandiflora Superba; and *TU*, Carolyn Tuttle.

### 3.3. Extraction Procedure

Heat-assisted extraction was carried out following a previously established protocol [10] with some modifications. Briefly, 2 g of lyophilized samples were weighed and placed into dark glass bottles. A total of 50 mL of solvent, a mixture of methanol (MeOH) and distilled water in a proportion of 60:40 (*v*/*v*), was then added. The extraction was conducted in a water bath at 50 °C for 1 h using constant magnetic stirring. Subsequently, the samples were collected in Falcon tubes and centrifuged at 4000× *g* for 15 min at room temperature. Finally, the supernatants were filtered using a 0.2 µm syringe filter and subjected to subsequent analyses. The extracts used for antimicrobial analysis were lyophilized and stored at −80 °C until analysis.

### 3.4. Phytochemical Characterization

Separation and detection of the phenolic compounds present in the EV–TU extracts obtained from the eight cultivars of the *C. japonica* flowers were performed by high-performance liquid chromatography coupled to a triple quadrupole tandem mass spectrometer via electrospray ionization (HPLC–ESI-QQQ-MS/MS, Thermo Scientific TSQ Quantis) in positive ion mode. HPLC separation was conducted using a Waters Spherisorb S3 ODS-2C_18_ column (150 × 3.9 mm, 4 µm particle) coupled to an in-line filter kit. Mobile phases A and B were water and acetonitrile, respectively, acidified with 0.1% formic acid (*v*/*v*). The HPLC gradient started with 100% A for 1 min, 95% A for 4 min, 85% A for 5 min, 70% A for 5 min, 40% for 5 min, 20% A for 5 min, and 5% A for 3 min; it was then switched back to 100% A for 2 min to ensure good HPLC column equilibration. The flow rate was 0.350 mL/min, and column temperature was maintained at 35 °C. The injection volume was 500 μL. Tandem mass analysis was performed after optimizing the tube lens and collision energy for each compound separately. On the other hand, the following ESI conditions were automatically adjusted to the set flow: nebulizing (N_2_) gas flow, 1.5 L/min; curved desolvation line and heat block temperatures, 200 °C; drying gas pressure, 100 kPa; and detector voltage, 1.65 kV. The profiling of phytochemicals was carried out using the comprehensive databases Phenol-Explorer3.6 (version) and MassBank and subsequently integrated with compounds obtained from the literature. The annotation process was conducted with a level two confidence, which entailed putative identification using the isotopic profile of each compound, ensuring a mass accuracy of less than 5 ppm. Data acquisition and HPLC–MS/MS analysis interpretation were conducted by means of Xcalibur software (version 4.1, Thermo Fisher Scientific, Waltham, MA, USA).

Following annotation, the chemical compounds were categorized into various phenolic classes and subclasses and then assessed, employing a semi-quantitative methodology. The quantification of each class was performed using a calibration curve established with pure analytical standards, all of which were of HPLC-grade quality and procured from Sigma-Aldrich (USA). The results were expressed as equivalents in micrograms (µg) of the representative compound per 100 g of dry weight (µg of equivalents/100 g dw). Thus, anthocyanin content was represented as cyanidin equivalents; flavanol content as (+)-catechin equivalents; flavone, isoflavonoids, and dihydrochalcone content as luteolin equivalents; flavonols and dihydroflavonols content as quercetin equivalents; hydroxybenzoic acids as gallic acid equivalents; hydroxycinnamic acids as ferulic acid equivalents; tyrosols and curcuminoids as tyrosol equivalents; and stilbene content as trans-resveratrol equivalents. The determinations were performed in quadruplicate (n = 4).

### 3.5. Antioxidant Capacity Determination

#### 3.5.1. DPPH Radical Scavenging Activity

This assay was founded on the oxidative stability and free radical release of α,α-diphenyl-β-picrylhydrazyl (DPPH) in the presence of an antioxidant-rich extract [85]. The DPPH RSA activity of the extracts of the *C. japonica* flowers was spectrophotometrically determined at 515 nm, against the radical DPPH as previously described [86]. Briefly, a volume of 50 μL of each *C. japonica* flower extract (serial dilutions using MeOH) was added to a microplate and mixed with 200 μL of DPPH methanol solution (190 mM). Once this mixture was homogenized, the absorbance was quantified (t = 0 min) immediately and after one hour (t = 60 min) using a wavelength of 515 nm in a Synergy HTX multi-mode reader (Bio-Tek, Winooski, VT, USA). The control of this assay consisted of 50 mL of MeOH and 200 μL of DPPH (190 mM). A calibration curve using a methanolic solution of Trolox (5–100 μg/mL) was generated to compare the scavenging activity ability of *C. japonica* flower extracts using the following Equation (1):(1)$$DPPH\left(\%\right)=\left(\frac{Ac-As}{Ac}\right)$$
where *Ac* is the absorbance of control at t = 0 and *As* is the absorbance of sample at t = 60. Absorbance data were expressed as the inhibitory percentage against the extract concentration (mg/mL). Data adjusted to a linear regression allowed for the determination of the half inhibitory concentration (IC_50_), which indicated the concentration of *C. japonica* flowers to scavenge half of DPPH radicals. Results were expressed as µg of Trolox equivalent (TE) per mL of extract. The determination was performed in quintuplicate and expressed as the mean ± standard deviation (SD) (n = 5).

#### 3.5.2. ABTS Radical Scavenging Activity

This assay was based on the quantification of the discoloration of the 2,2′-azino-bis(3-ethylbenzothiazoline-6-sulfonic acid (ABTS^•+^) cationic radical due to interaction with hydrogen or electron donor species [87]. The applied protocol was previously published [86]. Briefly, the activated ABTS^•+^ radical solution was prepared with 5 mL of 3.86 mg/mL of ABTS (Alfa Aesar, Thermo Fisher, Kandel, Germany) and 88 µL of a potassium persulfate solution at 37.8 mg/mL (Carlo Erba Reagents, Milan, Italy). This mixture was incubated overnight at room temperature and then diluted to 1:40 (*v*/*v*) using ethanol as solvent. Afterward, 200 µL of this activated ABTS^•+^ radical solution was mixed with 50 µL of each *C. japonica* flower extract (serial dilutions using ethanol) into a microplate. The absorbances of the mixtures were then measured spectrophotometrically at 734 nm (Synergy HTX multi-mode reader) at time zero (t = 0 min) and after 1 h of incubation at room temperature (t = 60 min). Ethanol was used as control. Trolox was used for the calibration curve, and the data analysis applied was the same as with DPPH. Thus, the inhibitory half concentration value (IC_50_, µg/mL), which refers to the *C. japonica* flowers extract concentration required to scavenge half of ABTS^•+^ radicals, was calculated. The determination was performed in quintuplicate and expressed as the mean ± standard deviation (SD) (n = 5).

#### 3.5.3. Crocin Bleaching Assay (CBA)

This method was based on the use of crocin as an oxidizable substrate and 2,2′-azobis 2-amidinopropane dihydrochloride (AAPH) as a source of free radicals. Through this assay, the competition established between crocin and the antioxidants present in the sample would be assessed. Therefore, the bleaching inhibition of crocin in the presence of *C. japonica* flowers was detected [88]. A solution of crocin (100 µmol/L) (Tokyo Chemical Industry C.O., Tokyo, Japan) and AAPH (7.68 mmol/L) (Acros organics, Morris Planes, NJ, USA) were added into 30 mL of preheated Milli-Q water (40 °C) with 100 mM of phosphate buffer (pH = 5.5), according to the methodology proposed by Prieto [88]. Thereafter, 50 μL of each *C. japonica* flower extract (serially diluted with Mili-Q water) was placed in a microplate with 200 μL of the previously prepared mixture of reagents. The absorbance was measured at 450 nm at time zero (t = 0 min) using a Synergy HTX multi-mode reader. Then, the samples were incubated on an orbital shaker (85/100 rpm) at 37 °C and the absorbance of each sample was regularly measured every 20 min until a total incubation time of 200 min. The control sample was prepared using a mixture of the above-mentioned reagents and Milli-Q water. The obtained data were adjusted to a kinetic profile based on the Weibull mass function, according to Equation (2) as suggested by Prieto [88].
(2)$${R=K\times (1-e}^{-\left(ln2\right)\times {\left(\frac{t}{\tau }\right)}^{\infty }})$$
where *K* is the asymptote, τ is the time when half of the oxidation is achieved, α is the shape parameter associated with the maximum slope of the response (v_max_). The assay was performed in quadruplicate and expressed as the mean ± standard deviation (SD) (n = 4).

### 3.6. Antioxidant Activity Determination

#### 3.6.1. Superoxide Radical Scavenging Activity (SRSA)

This assay was based on the generation superoxide radicals (O_2_^•−^) by applying non-enzymatic phenazine methosulfate–nicotinamide adenine dinucleotide (PMS/NADH). These superoxide radicals are meant to reduce nitro blue tetrazolium (NBT) into a purple formazan [89]. The determination of the superoxide radical scavenging activity was assayed according to a previously published method [90,91], with some modifications. Briefly, 50 μL of each *C. japonica* flower extract dissolved in phosphate buffer, ranking from 2.0 to 0.03125 mg/mL in concentration, were applied to a microplate. Then, 50 μL of NADH (166 μM), 50 μL of NBT (43 μM), and 50 μL of PMS (2.7 μM) were added to each well. Absorbance was recorded at 560 nm for 2 min using a microplate reader, Synergy^HT^ (BioTek Instruments). Ascorbic acid was used as positive control. The obtained data were adjusted to a kinetic profile based on the Weibull mass function, according to Equation (2), as suggested by Prieto et al. [88]. The results were expressed in mg/mL as the mean ± standard deviation (SD) (n = 4).

#### 3.6.2. Hydroxyl Radical Scavenging Activity (OHSA)

Hydroxyl radical scavenging activity was measured by the salicylic acid method [92], with some modifications. Briefly, the lyophilized extract was resuspended in deionized water at a concentration of 8 mg/mL, after which 6 serial dilutions were prepared up to the final concentration of 312.5 µg/mL. Then, 70 μL of this extract was transferred to a microplate and mixed with 70 μL of salicylic acid (9 mM), 70 μL of iron sulphate (9 mM), and 70 μL of hydrogen peroxide (H_2_O_2_, 9 mM). The reaction was carried out at 37 °C for 60 min [93]. The absorbance was then determined using a microplate reader, Synergy^HT^ (BioTek Instruments), at 510 nm. For the control group, the extract was replaced with distilled water. The blank sample did not contain H_2_O_2_, whereas ascorbic acid was used as the positive control. The obtained data were adjusted to a kinetic profile based on the Weibull mass function, according to Equation (2), as suggested by Prieto et al. [88]. Results were expressed as mg/mL (n = 3).

#### 3.6.3. Nitric Oxide Scavenging Assay (NOSA)

The nitric oxide scavenging assay was evaluated based on a diazotization reaction as described in previous works [90]. Briefly, *C. japonica* flower extracts were dissolved in a potassium phosphate buffer (0.1 M, pH = 7.4) at 20 mg/mL, and six serial dilutions were prepared. A volume of 100 µL of each *C. japonica* flower extract and 100 µL of sodium nitroprusside (20 mM) was added to each well, and the microplate was incubated at room temperature for 1 h. Then, 100 µL of Griess reagent (1% sulphanilamide and 0.1% naphthylethylenediamine in 2% phosphoric acid) was added to the mixture and left to stand for 10 min. After incubation, the absorbance of the mixtures was measured at 560 nm. The control group was made with potassium phosphate buffer instead of *C. japonica* flower extracts, while 2% phosphoric acid instead of Griess reagent was added to the blanks. Ascorbic acid was used as a positive control. The obtained data were adjusted to a kinetic profile based on the Weibull mass function, according to Equation (2), as suggested by Prieto et al. [88]. Results were expressed as mg/mL (n = 3).

#### 3.6.4. H_2_O_2_ Scavenging Assay

The scavenging activity of the *C. japonica* flower extracts towards H_2_O_2_ was assessed by measuring the signal attenuation at 230 nm [94]. For that, 75 µL of a 40 mM H_2_O_2_ solution was added to 425 µL of sample extract, previously dissolved in a 0.1 M potassium phosphate buffer (pH 7.4), at concentrations ranging from 27.2 to 870 µg/mL. The absorbance was measured after a 10 min incubation [95] using a Shimadzu UV-260 spectrophotometer. Ascorbic acid was employed as the positive control, while the buffer solution was used as the negative control. The scavenging activity of hydrogen peroxide was then calculated as previously explained, by adjusting data to a kinetic profile based on the Weibull mass function, according to Equation (2), as suggested by Prieto et al. [88]. Results were expressed as mg/mL (n = 3).

### 3.7. Antibacterial Tests

The determination of the antimicrobial activity was assayed according to the agar disk diffusion method, which consists of the cultivation of bacteria in a Petri dish and the subsequent measurement of inhibition zones produced by the addition of the extracts [96,97].

#### 3.7.1. Microorganisms and Culture Conditions

The antimicrobial activity of the different *C. japonica* flower extracts was assessed against the following Gram-positive bacterial strains: *S. aureus* (ATCC 25923), *S. epidermidis* (NCTC 11047), and *B. cereus* (ATCC 14579); and the following Gram-negative strains: *P. aeruginosa* (ATCC 10145), *S. enteritidis* (ATCC 13076), and *E. coli* (NCTC 9001). With the exception of *S. epidermidis*, an opportunistic bacteria [98], the microbes used are some of the most common food poisoning microorganisms [99]. The initial number of colony forming units was normalized (0.5 McFarland scale) by measuring the turbidity at 600 nm [100]. The lyophilized *C. japonica* flower extracts were dissolved in DMSO to the final concentration of 20 mg/mL and sterilized by filtration using a 0.22 µm syringe filter.

#### 3.7.2. Agar Diffusion Assay (ADA)

To determine the antimicrobial activity of the *C. japonica* flower extracts, the agar diffusion method [97] was adopted, with some modifications. Briefly, 50 μL of the previously cultivated inoculum were deposited in a Petri dish containing Mueller–Hinton Agar, where it was seeded and spread by streaking in 4 quadrants with sterile swabs. Next, 15 μL of dimethyl sulfoxide (DMSO) was added as negative control in one quadrant, 15 μL of lactic acid (40%) was added as positive control in another quadrant; and 15 μL of the extract (20 mg/mL in DMSO) was added in yet another quadrant [100]. The Petri dishes were incubated at 37 °C for 24 h and the inhibition zone diameters were determined with a digital caliper rule. The assay was conducted in triplicate, and the experimental data were expressed as the mean ± standard deviation (SD) (n = 3).

### 3.8. Cytotoxic Activity

The antiproliferative activity of the *C. japonica* flowers extracts was evaluated using the Sulforhodamine B (SRB) colorimetric assay, as previously described by Vichai [101]. The cytological models used in this study were the Vero, human gastric cancer (AGS), human hepatocellular carcinoma (HepG2), and human lung cancer (A549) cell lines. All these cell lines were provided by Frilabo (Porto, Portugal).

Briefly, 190 µL of cells at a concentration of 50,000 cells/mL were placed in a microplate and treated with 10 µL of each *C. japonica* flower extract. The starting concentration was 8 mg/mL and reached a minimum concentration of 31.25 µg/mL after several serial dilutions. The negative control consisted of 10 µL of non-supplemented medium. Plates were first incubated at room temperature for 1 h and then for 48 h at 37 °C with 5% CO_2_. Then, 100 µL of cold 10% trichloroacetic acid (Fisher Chemical Reagents, Pittsburgh, PA, USA) was added, and the plate was incubated at 4 °C for 1 h. After a washing step, another 100 µL of sulforhodamine B (0.057% *m*/*v* in 1% acetic acid) were added and incubated at room temperature for 30 min. After a final washing and drying process, 200 µL of 10 mM Tris were added and the plate was incubated at room temperature on an orbital shaker for 30 min. Absorbances were measured in a microplate reader at 540 nm. The determination of IC_50_ was performed in triplicate and expressed as the mean ± standard deviation (SD) (n = 3).

### 3.9. Neuroprotective Activity

The neuroprotective activity of the samples was evaluated using a previously developed colorimetric method, as described by Ellman [102], with minor modifications. This assay was based on the detection of the inhibition of the activity of the acetylcholinesterase (AChE) and butyrylcholinesterase (BuChE) enzymes, by evaluating the increase in yellow coloration as a result of the production of thiocholine [103]. Briefly, extracts were dissolved in Tris-HCl buffer (50 mM, pH = 8), and 25 μL of each mixture was added to the wells, along with 125 μL of 5,50-dithiobis(2-nitrobenzoic acid) reagent (DTNB), 50 μL of buffer B (Tris-HCl buffer + 0.1% albumin), 25 μL of acetylthiocholine iodide/S-butyrylthiocholine iodide (ATCI/BTCI), and 25 μL of 0.44 U/mL of AChE or 0.40 U/mL of BuChE solutions. The slopes were calculated from the kinetic curve obtained from six absorbance measurements at 405 nm, within a total reaction time of 1 min 44 s, after subtracting the blanks (wells containing all reagents except the enzyme). Galantamine was used as a positive control (effective half concentration value, EC_50_ = 0.92 g/mL against AChE and EC_50_ = 4.92 g/mL against BuChE) and Tris-HCl buffer as a negative control. The inhibitory activity values were calculated in accordance with Equation (3):(3)$$I\;(\%)=\frac{\left(AbsC-AbsCB\right)-(AbsM-AbsCB)}{\left(AbsC-AbsCB\right)}\times 100$$
where AbsC is the mean absorbance per minute of the control, AbsCB is the absorbance per minute of the control blank; and AbsM is the absorbance per minute of the sample. The determination was performed in triplicate and data were expressed as the mean ± standard deviation (SD) (n = 3).

### 3.10. Statistical Analysis

Data analysis was performed using R software version 4.2.0 (R Development Core Team, 2011). Before conducting parametric tests, the normality of the data distributions was assessed using the Shapiro–Wilk test. Analysis of variance (one-way ANOVA, *p* < 0.05) was employed to study the matrix effect on antioxidant capacity and activity of samples, provided that the data met the assumptions of normality and homogeneity of variances. Tukey–Kramer comparison tests were then conducted to study the matrix effect on antioxidant capacity and activity of the samples. To analyze the correlations between the different biological capacities and activities assessed, Pearson’s correlation analysis was performed using R software. The results were visualized as a correlation matrix in the form of a heatmap. A significance level of α = 0.01 was applied to determine statistically significant correlations. All statistical procedures were carried out ensuring that the assumptions for parametric testing were met, with normality verified prior to ANOVA and correlation analyses.

## 4. Conclusions

*C. japonica* flowers are a source of interesting natural bioactive compounds that could be employed in the development of new industrial applications. Our research was focused on the assessment of the antioxidant, antimicrobial, cytotoxic, anti-inflammatory, and neuroprotective activities of eight different cultivars. As expected, due to their similar composition, not all parameters were affected by the cultivar analyzed, showing a possible correlation between antioxidant capacity and the rest of bioactivities. In terms of antimicrobial activity, no cultivar showed effects against *E. coli*, with only three cultivars (DT, EM, and CT) showing activity against the rest of the microorganisms assessed. Among the evaluated cultivars, GS stands out for possessing a higher cytotoxic activity, with the HepG2 and A549 activity being approximately two times higher than the AGS activity. None of the cultivars under study had promising neuroprotective effects. Therefore, the flowers of *C. japonica* can be considered a potential source of various pharmacological compounds against microorganisms or tumors. However, though the outcome of the study and the bibliographic resources seem to be very promising, there are still certain obstacles being faced in developing these extracts into a drug. An interdisciplinary approach of various fields is needed such that a chemical modification strategy or nano-drug delivery mechanisms can be introduced to enhance its activity.

## Figures and Tables

**Figure 1 pharmaceuticals-17-00946-f001:**
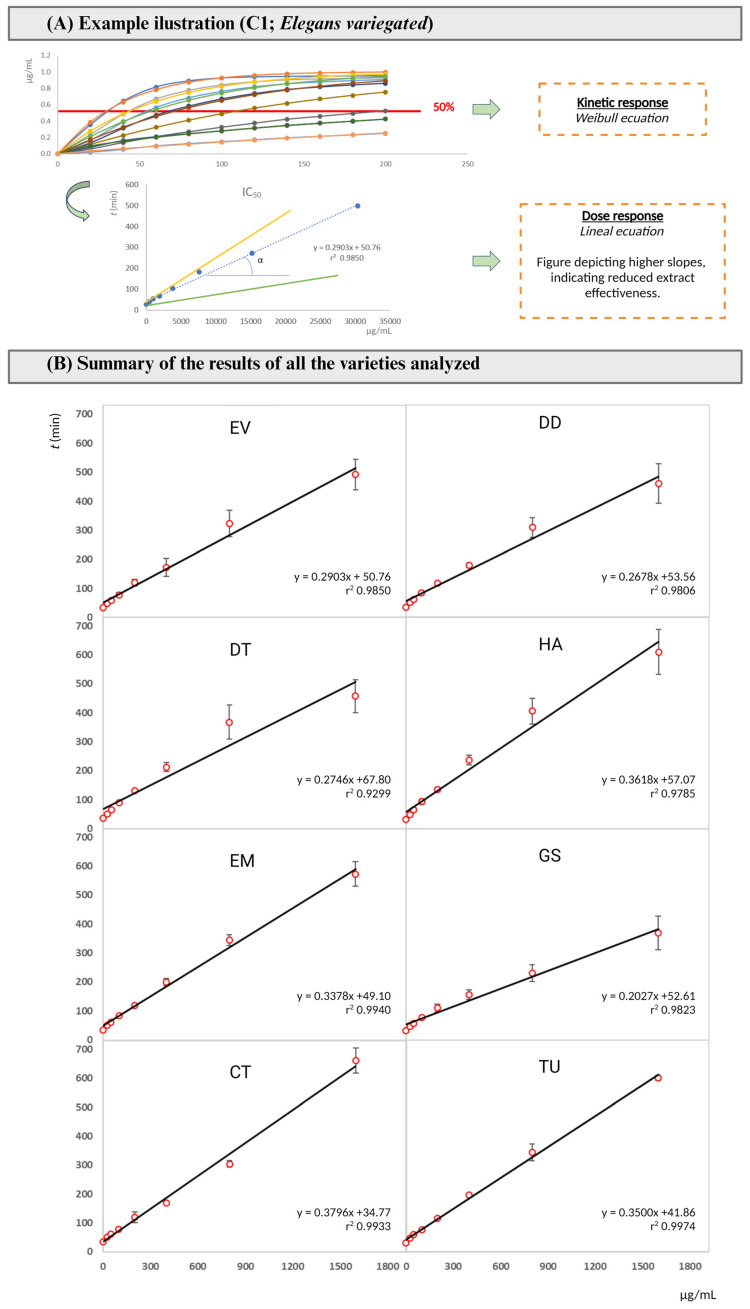
Results of crocin assay. (**A**) Raw experimental signal for the Elegans Variegated cultivarcultivar, with the experimental data fitted to Equation (2) to determine IC_50_ values. (**B**) Compilation of IC_50_ values for all cultivars, providing a comparative view of crocin assay outcomes across the different *C. japonica* flower cultivars.

**Figure 2 pharmaceuticals-17-00946-f002:**
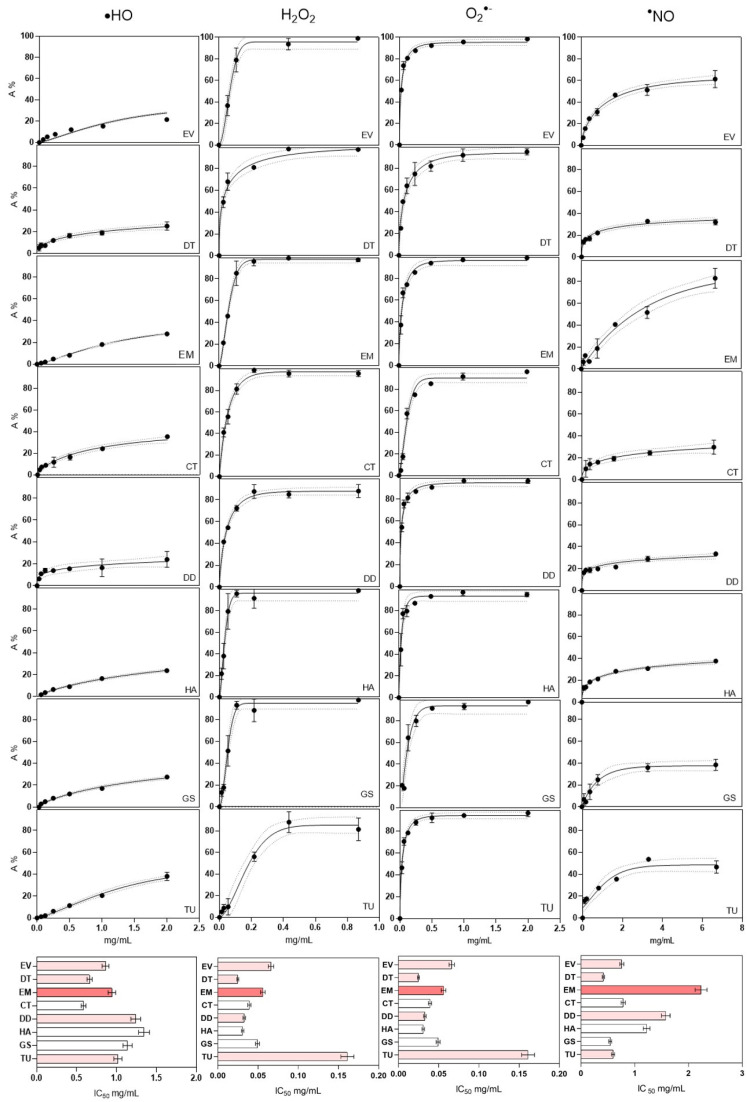
Graphs of in vitro antioxidant studies. Column 1 illustrates hydroxyl radical scavenging activity, column 2 showcases H_2_O_2_ scavenging activity, column 3 demonstrates superoxide radical scavenging activity, and column 4 displays nitric oxide scavenging activity. Symbols represent experimental data, while lines depict model values. The bar graphs in the final row present the IC_50_ values (mg/mL) for each assay across all the cultivars examined in this study.

**Figure 3 pharmaceuticals-17-00946-f003:**
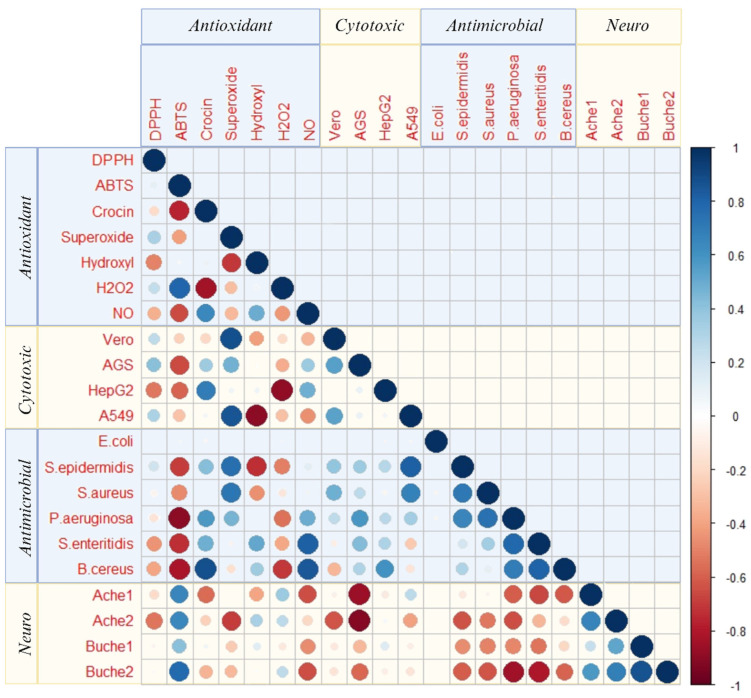
Pearson’s correlation heatmap between the different biological activities of *C. japonica* L. flower extracts (antioxidant activity, by means of DPPH, ABTS, CBA, SRSA, OHSA, NOSA, H_2_O_2_ methods; cytotoxic activity towards Vero, AGS, HepG2, A549 cell lines; anti-microbial activity against *E. coli*, *S. epidermidis*, *S. aureus*, *P. aeruginosa*, *S. enteritidis*, and *B. cereus*; and neuroprotective activity).

**Table 1 pharmaceuticals-17-00946-t001:** Compounds annotated in *C. japonica* flowers using HPLC–MS-MS in positive mode.

ID	Class	Compound	Formula	Precursor	Products	Col.	RF	Rt	Flowers of *C. japonica* Cultivars							
									*EV*	*DT*	*EM*	*CT*	*DD*	*HA*	*GS*	*TU*
				(*m/z*)	(*m/z*)	(V)	(V)	(min)	(μg/100 g Extract)							
*P1 *	*Anthocyanins *	Cyanidin 3-*O*-arabinoside	C_20_H_19_ClO_10_	449.233	287.083	20	107	4.6	0.098	0.171	0.192	0.144	0.007	0.268	0.225	0.145
*P2 *		Pelargonidin-3-*O*-rutinoside (isomer)	C_27_H_31_O_14_^+^	579.383	289.167/291.167/409.167/427.25	15–20	188	6.19	0.073	0.058	0.098	0.093	0.052	0.088	0.065	0.053
*P3 *		Pelargonidin-3-*O*-rutinoside (isomer)	C_27_H_31_O_14_^+^	579.383	289.167/291.167/409.167/427.25	15–20	188	6.41	0.043	0.039	0.043	0.032	0.054	0.040	0.035	0.001
*P4 *		Pelargonidin 3-*O*-arabinoside	C_20_H_19_O_9_^+^	402.833	258.167/313.25	10	110	9.71	0.001	0.001	0.001	0.001	0.001	0.001	0.001	0.001
*P5 *		Pelargonidin-3-*O*-sambubioside	C_26_H_29_O_14_^+^	579.333	222.667/408.167	15	163	12.63	0.001	0.001	0.001	0.001	0.001	0.001	0.001	0.001
*P6 *		Peonidin	C_16_H_13_O_6_^+^	304.43	178.083/219/260.25	5–15	93	15.25	3.291	3.132	12.256	15.965	2.118	4.051	1.348	1.359
*P7 *		Peonidin derivate	C_27_H_31_O_15_^+^	579.383	287.083	28	171	22.12	1.548	1.496	7.017	1.508	1.489	1.531	1.472	1.525
*P8 *		Malvidin-3-*O*-glucoside	C_23_H_25_O_12_^+^	449.2	258.167/291/373.167	12–17	107	16.61	0.655	0.613	6.364	1.069	0.521	0.575	0.588	0.638
*P9 *		Malvidin 3-*O*-(6″-caffeoyl-glucoside)	C_32_H_31_O_15_	671.317	365.167/475.25/565.333	11–14	128	18.68	1.458	1.424	5.128	1.421	1.644	1.474	1.374	1.472
*P10 *		Cyanidin 3-*O*-glucosyl-rutinoside	C_33_H_41_O_20_^+^	717.617	303.25/577.25	15–24	166	19.75	0.960	0.928	4.475	0.915	0.916	0.956	0.963	1.007
*P11 *	*Curcuminoids *	Curcumin	C_21_H_20_O_6_	381.15	204.917/277.083/349.167	9 -21	104	12.47	0.001	0.001	0.021	0.001	0.013	0.002	0.005	0.001
*P12 *	*Dihydrochalcones *	3-hydroxyphloretin-2-*O*-glucoside	C_21_H_24_O_11_	584.5	405/423.083/564.5	8–15	120	13.51	0.002	0.003	0.013	0.001	0.004	0.002	0.003	0.002
*P13 *		3-hydroxyphloretin 2-*O*-xylosyl-glucoside	C_26_H_32_O_15_	579.3	405.167/564.583	5–15	124	13.7	0.005	0.004	0.030	0.005	0.004	0.005	0.006	0.006
*P14 *		Phloridzin	C_21_H_24_O_10_	449.317	200/255/279.083/309.083	14–18	91	18.25	0.410	0.367	1.163	0.414	0.335	0.363	0.411	0.352
*P15 *	*Dihydroflavonols *	Dihydroquercetin (isomer 1)	C_15_H_12_O_7_	304.433	58.083/91.083/212.333	21–30	145	14.82	8.895	8.571	1.404	22.652	0.055	17.573	4.622	5.024
*P16 *		Dihydroquercetin (isomer 2)	C_15_H_12_O_7_	304.433	58.083/91.083/212.333	21–30	145	14.39	0.119	0.045	0.352	34.284	0.043	33.313	0.109	0.095
*P17 *		Dihydroquercetin (isomer 3)	C_15_H_12_O_7_	304.433	58.083/91.083/212.333	21–30	145	14.6	13.351	0.151	0.376	28.479	0.059	27.941	0.165	0.100
*P18 *		Dihydroquercetin (isomer 4)	C_15_H_12_O_7_	304.433	58.083/91.083/212.333	21–30	145	17.19	0.674	0.658	2.993	0.562	0.418	0.459	0.630	0.666
*P19 *	*Flavonols *	Quercetin-3-*O*-arabinose	C_20_H_18_O_11_	449.15	254.667/302.833/309.167	12–19	101	15.46	1.386	1.352	7.854	9.322	1.138	1.265	0.634	0.592
*P20 *		Kaempferol 3-*O*-acetyl-glucoside	C_23_H_22_O_12_	449.267	148.167/200.083/255/399.667	10–19	99	18.04	2.075	1.860	9.890	1.843	1.532	1.846	1.862	1.886
*P21 *	*Flavone *	Nobiletin	C_21_H_22_O_8_	449.233	245.167/369	10–21	114	15.99	0.478	0.100	0.642	1.583	0.185	1.337	0.085	0.077
*P22 *		Apigenin 7-*O*-glucoside	C_21_H_20_O_10_	453.583	171.25/249/379.083	6–26	101	17.31	0.138	0.129	0.557	0.104	0.082	0.089	0.129	0.135
*P23 *		Apigenin 7-*O*-glucuronide	C_21_H_18_O_11_	449.2	326.75/365.333/406	6–18	91	21.98	0.230	0.219	1.044	0.214	0.214	0.224	0.215	0.222
*P24 *	*Hydroxybenzoic acids *	Ellagic acid acetyl-arabinose	C_21_H_16_O_13_	449.233	134	15	112	17.05	2.360	2.177	14.208	2.553	1.624	1.753	2.105	2.216
*P25 *		Gallic acid 4-*O*-glucoside	C_13_H_16_O_10_	332.3	91.083/240.333	22–31	152	25.77	20.838	21.114	52.515	20.207	20.826	21.787	21.159	22.521
*P26 *	*Hydroxycinnamic acids *	*p*-coumaroylquinic acid	C_16_H_18_O_8_	381.267	173/219.083/249/298	5 -27	85	16.36	0.420	0.439	1.311	1.363	0.360	0.474	0.320	0.380
*P27 *		Rosmarinic acid	C_18_H_16_O_8_	332.517	105–319	5–32	87	19.12	0.448	0.446	1.867	0.439	0.465	0.450	0.431	0.444
*P28 *		3,4-dicaffeoylquinic acid	C_25_H_24_O_12_	503.35	217.083/337.167	13–31	91	20.4	0.552	0.509	2.104	0.488	0.495	0.539	0.546	0.575
*P29 *	*Isoflavonoids *	6-*O*-malonylglycitin	C_25_H_24_O_13_	532.1	337.083/351.083	14	113	13.9	0.007	0.005	0.042	0.006	0.007	0.005	0.006	0.007
*P30 *		6-*O*-acetyldaidzin	C_23_H_22_O_10_	449.233	115/367.333/381.083	12–16	157	15.03	1.009	0.918	2.032	3.486	1.904	1.773	0.445	0.616
*P31 *		Daidzin	C_21_H_20_O_9_	449.233	295.083/325	11–18	128	15.78	0.137	0.135	0.939	2.477	0.126	2.325	0.090	0.092
*P32 *		Glycitin	C_22_H_22_O_10_	449.2	257.167/267	20–24	96	16.6	0.115	0.111	1.433	0.270	0.103	0.100	0.092	0.100
*P33 *		6-*O*-malonyldaidzin	C_24_H_22_O_12_	503.283	308.833/323/471.167	16–28	114	20.61	0.189	0.177	0.923	0.179	0.177	0.180	0.174	0.176
*P34 *	*Stilbenes *	Pallidol	C_28_H_22_O_6_	449.1333	112.917	15	115	16.17	0.159	0.118	0.949	0.144	0.148	0.105	0.127	0.167
*P35 *	*Tyrosols *	Oleuropein	C_25_H_32_O_13_	579.3	361.167/379.25/407.25	9–16	118	12.84	22.900	20.833	46.113	79.092	43.211	40.230	10.109	13.968
*P36 *		Demethyloleuropein	C_24_H_30_O_13_	503.497	347.083/365.083	15–29	105	18.47	3.103	3.073	21.304	56.205	2.857	52.751	2.045	2.087

ID, code of the identified compound; Col., collision energy; RF, RF lens; Rt; retention time; V, volts; min, minutes; *m/z*; mass–charge ratio; *EV*, Elegans variegated; *DT*, Dr Tinsley; *EM*, Eugenia de Montijo; *CT*, Conde de la Torre; *DD*, Donation dentada; *HA*, Hagoromo; *GS*, Grandiflora Superba; *TU*, Carolyn Tuttle. *Note*: all values presented in the table exhibit a standard deviation consistently below 5%.

**Table 2 pharmaceuticals-17-00946-t002:** Bioactive properties of the methanolic extract of the 8 flowers cultivars of *C. japonica*.

	Control	EV	DT	EM	CT	DD	HA	GS	TU	*p*-Value
(*A*) *ANTIOXIDANT CAPACITY*										
DPPH (μg TE/mL)	-	35 ^^abc^^ ± 2	33 ^^bc^^ ± 2	30 ^c^ ± 2	39 ^a^ ± 1	37 ^ab^ ± 3	30 ^c^ ± 4	30 ^c^ ± 1	35 ^abc^ ± 4	2.7 × 10^−4^
ABTS (μg TE/mL)	-	121 ^a^ ± 9	88 ^ab^ ± 21	81 ^b^ ± 19	77 ^b^ ± 7	80 ^b^ ± 6	86 ^b^ ± 11	96 ^ab^ ± 11	121 ^a^ ± 18	5.1 × 10^−4^
Crocin (min/g dw)	-	0.2 ^b^ ± 0.2	0.27 ^ab^ ± 0.06	0.30 ^ab^ ± 0.08	0.31 ^ab^ ± 0.02	0.23 ^b^ ± 0.05	0.43 ^a^ ± 0.07	0.01 ^c^ ± 0.01	0.02 ^c^ ± 0.01	1.5 × 10^−7^
(*B*) *ANTIOXIDANT ACTIVITY*										
Superoxide (mg/mL)	-	0.022 ^a^ ± 0.001	0.0670 ^b^ ± 0.0033	0.040 ^c^ ± 0.002	0.1120 ^d^ ± 0.0056	0.0180 ^a^ ± 0.0009	0.029 ^a^ ± 0.001	0.096 ^e^ ± 0.005	0.028 ^a^ ± 0.028	1 × 10^−4^
Hydroxyl (mg/mL)	-	0.862 ^a^ ± 0.043	0.662 ^b^ ± 0.031	0.942 ^a^ ± 0.047	0.587 ^b^ ± 0.029	1.240 ^c^ ± 0.062	1.344 ^c^ ± 0.067	1.138 ^cd^ ± 0.057	1.017 ^cd^ ± 0.051	1 × 10^−4^
H_2_O_2_ (mg/mL)	-	0.0660 ^a^ ± 0.0033	0.0250 ^b^ ± 0.0012	0.0560 ^ac^ ± 0.0028	0.039 ^d^ ± 0.0019	0.0330 ^bd^ ± 0.0016	0.031 ^bd^ ± 0.0015	0.049 ^cd^ ± 0.002	0.161 ^e^ ± 0.0081	1 × 10^−4^
NO (mg/mL)	1.08	0.760 ^a^ ± 0.038	0.417 ^b^ ± 0.021	2.23 ^c^ ± 0.11	0.785 ^a^ ± 0.039	1.578 ^c^ ± 0.078	1.221 ^c^ ± 0.061	0.546 ^b^ ± 0.027	0.597 ^ab^ ± 0.029	1 × 10^−4^
(*C*) *ANTIMICROBIAL ACTIVITY* (mm)										
*E. coli*	17.7	nd	nd	nd	nd	nd	nd	nd	nd	-
*S. epidermidis*	24.4	nd	11 ^a^ ± 2	11 ^a^ ± 3	14 ^a^ ± 1	nd	nd	nd	nd	0.02
*S. aureus*	19.0	nd	11.0 ^a^ ± 0.8	9.7 ^a^ ± 0.6	11 ^a^ ± 1	nd	9.7 ^a^ ± 0.1	9.4 ^a^ ± 0.6	10.3 ^a^ ± 0.5	1.9 × 10^−13^
*P. aeruginosa*	19.2	7 ^b^ ± 3	10.0 ^ab^ ± 0.8	9.8 ^ab^ ± 0.6	10.4 ^a^ ± 0.7	9.2 ^ab^ ± 0.4	10.8 ^a^ ± 0.8	9.3 ^ab^ ± 0.7	9.2 ^ab^ ± 0.5	0.04
*S. enteritidis*	18.7	nd	6 ^a^ ± 2	10 ^a^ ± 1	8 ^a^ ± 2	10.5 ^a^ ± 0.3	11 ^a^ ± 1	8.4 ^a^ ± 0.7	7 ^a^ ± 1	1.2 × 10^−4^
*B. cereus*	17.4	nd	6.5 ^ab^ ± 0.4	7 ^a^ ± 1	3 ^b^ ± 3	9 ^a^ ± 1	9.4 ^a^ ± 0.7	nd	nd	1.9 × 10^−8^
(*D*) *CYTOTOXIC ACTIVITY* (μg/mL)										
Vero	9.0	18 ^b^ ± 1	9 ^b^ ± 8	18 ^b^ ± 1	52 ^a^ ± 25	6.23 ^b^ ± 0.03	17 ^b^ ± 11	69 ^a^ ± 12	9 ^b^ ± 2	1.7 × 10^−5^
AGS	3.7	22 ^ab^ ± 5	9 ^b^ ± 8	27 ^ab^ ± 5	52 ^a^ ± 25	36 ^ab^ ± 7	34 ^ab^ ± 8	32 ^ab^ ± 16	23 ^ab^ ± 1	0.02
HepG2	3.9	40 ^a^ ± 7	38 ^a^ ± 20	45 ^a^ ± 13	33 ^a^ ± 17	35 ^a^ ± 5	39 ^a^ ± 3	41 ^a^ ± 10	13 ^a^ ± 4	0.11
A549	1.9	15.0 ^ab^ ± 0.3	1.65 ^a^ ± 0.08	12.4 ^ab^ ± 0.9	22 ^ab^ ± 2	13.40 ^b^ ± 0.05	10.56 ^ab^ ± 0.4	19 ^ab^ ± 3	13.3 ^ab^ ± 0.6	0.02
(*E*) *NEUROPROTECTIVE ACTIVITY* (g/mL)										
AchE (%) 1 mg/mL	0.92	20 ^ab^ ± 3	25 ^a^ ± 2	16 ^ab^ ± 2	8 ^b^ ± 1	10 ^b^ ± 3	7 ^b^ ± 1	20 ^ab^ ± 6	16 ^ab^ ± 8	2.1 × 10^−3^
2 mg/mL		25 ^ab^ ± 2	26 ^a^ ± 7	20 ^ab^ ± 5	9 ^b^ ± 3	19 ^ab^ ± 5	22 ^ab^ ± 6	20 ^ab^ ± 8	21 ^ab^ ± 3	0.09
BuChE (%) 1 mg/mL	4.92	33 ^ab^ ± 5	25 ^abc^ ± 7	4 ^c^ ± 2	20 ^abc^ ± 6	16 ^bc^ ± 8	37 ^a^ ± 7	20 ^abc^ ± 4	14 ^c^ ± 6	5.8 × 10^−4^
2 mg/mL		47 ^a^ ± 2	29 ^ab^ ± 16	11 ^b^ ± 3	20 ^b^ ± 8	21 ^b^ ± 2	31 ^ab^ ± 4	28 ^ab^ ± 5	23 ^b^ ± 2	3.6 × 10^−3^

*EV*, Elegans variegated; *DT*, Dr Tinsley; *EM*, Eugenia de Montijo; *CT*, Conde de la Torre; *DD*, Donation dentada; *HA*, Hagoromo; *GS*, Grandiflora Superba; *TU*, Carolyn Tuttle. TE, trolox equivalents; dw, dry weight. DPPH, DPPH radical scavenging activity; ABTS, ABTS radical scavenging activity; Crocin, crocin bleaching assay; Superoxide, superoxide radical scavenging activity; Hydroxyl, hydroxyl radical scavenging activity; H_2_O_2_, hydrogen peroxide scavenging activity; NO, nitric oxide scavenging assay; *S. aureus*, *Staphylococcus aureus* (ATCC 25923); *S. epidermidis*, *Staphylococcus epidermidis* (NCTC 11047); *B. cereus*, *Bacillus cereus* (ATCC 14579); *P. aeruginosa*, *Pseudomonas aeruginosa* (ATCC 10145); *S. enteritidis*, *Salmonella enteritidis* (ATCC 13076); *E. coli, Escherichia coli* (NCTC 9001). Vero, Vero cell line; AGS, human gastric cancer cell line; HepG2, human hepatocellular carcinoma; A549, human lung cancer; AChE, inhibition of the activity of acetylcholinesterase enzyme; BuChE, inhibition of the activity of butyrylcholinesterase enzyme; Controls. Antioxidant activity: ascorbic acid and trolox; Antimicrobial activity: lactic acid; cytotoxic activity: ellipticine; Neuroprotective activity: galantamine as positive control and Tris-HCl buffer as a negative control; nd, not detected. Different letters within the same column indicate statistically significant differences (α = 0.05), according to one-way ANOVA.

## Data Availability

All related data are presented in this paper. Additional inquiries should be addressed to the corresponding author.

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
