# Peer review of "Unraveling the Bioactive Potential of Camellia japonica Edible Flowers: Profiling Antioxidant Substances and In Vitro Bioactivity Assessment"

_pharmaceuticals, 2024, doi:10.3390/ph17070946_

Round 1

Reviewer 1 Report

Comments and Suggestions for Authors

Change the title, why only phenolic mentioned? Can say antioxidant substances.

Make Abbreviations list

What is P1 to P 36?

Why was it not done antioxidant enzymes?

Rewrite the Statistical Analysis. What the experimental design and how many replicates? 

Comments on the Quality of English Language

Minor revision for english language

Author Response

Change the title, why only phenolic mentioned? Can say antioxidant substances.

Answer: Thank you for your valuable feedback on our manuscript. We have taken your suggestion into consideration and have changed the title to include "antioxidant substances" instead of just "phenolic."

The revised title is now: “Unraveling the bioactive potential of Camellia japonica edible flowers: profiling antioxidant substances and in vitro bioactivity assessment.”

We believe this new title better reflects the scope of our study. Thank you again for your insightful suggestion.

Make Abbreviations list

Answer: Thank you for your suggestion regarding the inclusion of an Abbreviations list in our manuscript. We have implemented this change and have added a comprehensive Abbreviations list at the beginning of the manuscript to enhance the clarity and readability of the manuscript.

What is P1 to P 36?

Answer: They are the ID, the code of the identified compounds. It has been clarified at the bottom of the table to avoid confusion.

Why was it not done antioxidant enzymes?

Answer: Thank you for your insightful question regarding the inclusion of antioxidant enzyme analysis in our study. We chose to focus on the compounds profiling and in vitro bioactivity assessment of Camellia japonica edible flowers due to the significant bioactive properties and antioxidant potential of phenolic compounds. Our primary aim was to characterize these compounds and evaluate their bioactivity.

However, we acknowledge the importance of antioxidant enzymes in the overall antioxidant capacity. Due to resource and time constraints, we were unable to include this analysis in the current study. We recognize that investigating antioxidant enzymes would provide a more comprehensive understanding of the antioxidant mechanisms, and we plan to address this in future research. We appreciate your feedback and will consider it for the expansion of our study.

Rewrite the Statistical Analysis. What is the experimental design and how many replicates?
Answer: We have ensured that all details about the experimental design and the number of replicates used for each technique are clearly explained in the respective sections of the "Materials and Methods". The updated "Statistical Analysis" section is now as follows:

“Data analysis was performed using R software version 4.2.0 (R Development Core Team, 2011). Before conducting parametric tests, the normality of data distributions was assessed using the Shapiro-Wilk test. Analysis of variance (one-way ANOVA, p < 0.05) was employed to study the matrix effect on antioxidant capacity and activity of samples, provided that the data met the assumptions of normality and homogeneity of variances. Tukey-Kramer comparison tests were then conducted to study the matrix effect on the antioxidant capacity and activity of samples. To analyze correlations between different biological capacities and activities assessed, Pearson’s correlation analysis was performed using R software. The results were visualized as a correlation matrix in the form of a heatmap. A significance level of α = 0.01 was applied to determine statistically significant correlations. All statistical procedures were carried out ensuring that the assumptions for parametric testing were met, with normality verified prior to ANOVA and correlation analyses.”

Thank you again for your insightful comments, which have helped to improve the clarity and rigor of our manuscript.

Reviewer 2 Report

Comments and Suggestions for Authors

This manuscript reports on the phytochemical study of the flowers of 8 cultivars of Camelia japonica and the study of the antioxidant, antibacterial and neuroprotective properties. The results are of interest, however there is one important point which needs additional attention by the Authors: 

They should explain how they performed compound identification using the dat from HPLC-ESI-MS/MS analysis. Did they use reference compounds? Did they use published data?

Please clarify this  and give degrees of confidence of the positive identification.

Author Response

This manuscript reports on the phytochemical study of the flowers of 8 cultivars of Camelia japonica and the study of the antioxidant, antibacterial and neuroprotective properties. The results are of interest, however there is one important point which needs additional attention by the Authors:

They should explain how they performed compound identification using the data from HPLC-ESI-MS/MS analysis. Did they use reference compounds? Did they use published data? Please clarify this and give degrees of confidence of the positive identification.

Answer: Thank you for your valuable feedback and for highlighting the need for further clarification on the compound identification process using HPLC-ESI-MS/MS analysis.

We performed the compound identification by comparing the retention times, mass spectra, and fragmentation patterns with those of reference compounds when available. For compounds where reference standards were not available, we relied on published data and databases to identify and confirm the compounds. Additionally, many of these compounds had been previously identified in metabolomics studies on Camellia species, which further supported our identification process.

We have updated the manuscript to include these details and the degrees of confidence for the identified compounds. Thank you for your constructive comments, which have helped improve the clarity of our methodology.

Reviewer 3 Report

Comments and Suggestions for Authors

The revised manuscript “Unraveling the bioactive potential of Camellia japonica edible flowers: phenolic profiling and in vitro bioactivity assessment by Pereira et al.” is an original, well-developed and correctly presented work related to the phytochemical composition and biological activities of 8 different varieties of C. japonica flowers.   

The sections of the manuscript are precise and ordered. Into the introduction, all the relevant antecedents are presented and the aim of the work clearly expressed. In the Material and methods section, the information is complete and valuable to the result interpretation. The results presentation is unified with the discussion, which is a convenient decision to this type of manuscript. The overall section is ordered and display clearly the results obtained, interpreting its contribution to the previous state of the knowledge. The arrived conclusions are supported by the results and it is coherent with the rest of the manuscript. The tables are well presented and contribute to the results valuation. Finally, the cited bibliography is complete, pertinent and includes recent cites.

In conclusion, in accordance to this reviewer criteria the manuscript is able to be published in Pharmaceuticals Journal.

Author Response

The revised manuscript “Unraveling the bioactive potential of Camellia japonica edible flowers: phenolic profiling and in vitro bioactivity assessment by Pereira et al.” is an original, well-developed and correctly presented work related to the phytochemical composition and biological activities of 8 different varieties of C. japonica flowers.

The sections of the manuscript are precise and ordered. Into the introduction, all the relevant antecedents are presented and the aim of the work clearly expressed. In the Material and methods section, the information is complete and valuable to the result interpretation. The results presentation is unified with the discussion, which is a convenient decision to this type of manuscript. The overall section is ordered and display clearly the results obtained, interpreting its contribution to the previous state of the knowledge. The arrived conclusions are supported by the results and it is coherent with the rest of the manuscript. The tables are well presented and contribute to the results valuation. Finally, the cited bibliography is complete, pertinent and includes recent cites.

In conclusion, in accordance to this reviewer criteria the manuscript is able to be published in Pharmaceuticals Journal.

Answer: Thank you very much for your thorough and positive review of our manuscript. We are delighted to hear that you found our work original, well-developed, and correctly presented.

We appreciate your comments on the precision and organization of the sections, as well as the completeness and value of the information provided in the Materials and Methods section. Your recognition of the unified presentation of results and discussion, and its contribution to the manuscript's clarity, is especially gratifying. We are also pleased that you found the conclusions to be well-supported by the results and coherent with the rest of the manuscript, and that the tables and bibliography were well-presented and pertinent.

Thank you again for your kind words and for recommending our manuscript for publication in Pharmaceuticals Journal.

Reviewer 4 Report

Comments and Suggestions for Authors

The manuscript pharmaceuticals-3057867 describes the LC-MS-based chemical characterization of flowers from eight C. japonica plant variations. The manuscript has important elements and could be interesting for readers. However, some issues should be addressed before further consideration.

1. Line 29: Some specific results can be provided to support the next conclusive sentence.

2. Line 120: Are the test C. japonica flowers related to varieties or cultivars? Revise them. In addition, are white and pink cultivars related to phenotypes instead of ecotypes?

3. Line 123: Provide the flowers' status, age, and conditions at the collection moment.

4. Line 126: How many flowers per plant per cultivar were collected?  Specify it. Information about biological and technical replicates is also missing.

5. Line 136: How many flowers were lyophilized to get the 2 g of material? Specify it.

6. Line 144 and 356: Why bioactive? Sections 2.4 and 3.1 are merely related to chemical characterization. The term bioactive should always be linked to in-study experimental evidence. Through LC-MS, it is not possible to determine the bioactivity of a target compound or to know if a compound is bioactive since it is only related to a chemical characterization based on the MS data and the comparison with standards, as the study mentioned.

7. Lines 174-211: There is a difference between "antioxidant capacity" (a thermodynamic assessment of radical scavenging assays) and "antioxidant activity" (a kinetic evaluation), and this study involves both. The manuscript should refer to the correct term when appropriate. Be consistent throughout the manuscript.

8. Line 349: Was the data normality verified to support a parametric test? Revise and add the relevant information to the manuscript.

9. Table 1: How were the dihydroquercetin isomers quantified in ug/g extract if the correct isomer was not determined? Which standard was used for such quantifications if different per isomer per cultivar? These explanations must be clearly added to the manuscript to avoid confusing interpretations.

Comments on the Quality of English Language

Some grammar and stylistic issues must be deeply revised. Detailed scrutiny should be performed throughout the manuscript to revise such issues.

Author Response

The manuscript pharmaceuticals-3057867 describes the LC-MS-based chemical characterization of flowers from eight C. japonica plant variations. The manuscript has important elements and could be interesting for readers. However, some issues should be addressed before further consideration.

  1. Line 29: Some specific results can be provided to support the next conclusive sentence.

Answer: Thank you for your valuable feedback. We have carefully reviewed your suggestion and incorporated specific results to support the conclusive sentence in the revised manuscript. By adding detailed correlation coefficients for each bioactivity tested, we aim to provide a clearer and more robust demonstration of the significant relationship between the phenolic compounds and their bioactivities.

  1. Line 120: Are the test C. japonica flowers related to varieties or cultivars? Revise them. In addition, are white and pink cultivars related to phenotypes instead of ecotypes?

Answer: Thank you for your insightful comments regarding the classification of the test C. japonica flowers. Upon reviewing the manuscript, I acknowledge that you are correct. According to the International Union for the Protection of New Varieties of Plants (UPOV) and various articles, the appropriate term is indeed "cultivar." Additionally, I have reviewed and revised the definitions, particularly distinguishing between phenotypes and ecotypes. The manuscript has been updated accordingly to reflect these corrections. Thank you for your valuable feedback.

  1. Line 123: Provide the flowers' status, age, and conditions at the collection moment.

Answer: We have added the requested information regarding the status, age, and conditions of the flowers at the time of collection. The updated manuscript now includes the following details:

“Studies were conducted using eight different cultivars of 20-year-old C. japonica flowers, 4 of them of white phenotype (Conde de la Torre, Dr Tinsley, Grandiflora Superba, Hagoromo) and 4 of pink phenotype (Elegans variegated, Donation dentada, Eugenia de Montijo and Carolyn Tuttle). All of them were manually harvested in NW Spain (42.431º N, 8.6444º W) during January 2020 at their optimal state of maturity (stage 5, full bloom), harvesting a total quantity of 400 g of fresh flowers (around 15 flowers for each cultivar).”

  1. Line 126: How many flowers per plant per cultivar were collected? Specify it. Information about biological and technical replicates is also missing.

Answer: We have added the requested information regarding the number of flowers and biological and technical replicates. The updated manuscript now includes the following details:

“Studies were conducted using eight different cultivars of 20-year-old C. japonica flowers, 4 of them of white phenotype (Conde de la Torre, Dr Tinsley, Grandiflora Superba, Hagoromo) and 4 of pink phenotype (Elegans variegated, Donation dentada, Eugenia de Montijo and Carolyn Tuttle). All of them were manually harvested in NW Spain (42.431º N, 8.6444º W) during January 2020 at their optimal state of maturity, harvesting a total quantity of 400 g of fresh flowers (around 15 flowers for each cultivar).”

  1. Line 136: How many flowers were lyophilized to get the 2 g of material? Specify it.

Answer: In response to your question regarding the quantity of flowers lyophilized to obtain the 2 g of material, we have made the following additions and clarifications in the manuscript:

We harvested a total of 400 g of fresh flowers, which corresponds to approximately 15 flowers for each cultivar. It is important to note that the fresh flowers have a high moisture content, reaching up to 80%. This high moisture content significantly reduces the weight of the flowers once lyophilized. Consequently, the lyophilization process yielded more than the 2 g of dry material required for the assay.

This additional information has been incorporated into the manuscript to provide a clearer understanding of the material preparation process. We believe this addresses your concern and enhances the transparency and reproducibility of our experimental procedure.

  1. Line 144 and 356: Why bioactive? Sections 2.4 and 3.1 are merely related to chemical characterization. The term bioactive should always be linked to in-study experimental evidence. Through LC-MS, it is not possible to determine the bioactivity of a target compound or to know if a compound is bioactive since it is only related to a chemical characterization based on the MS data and the comparison with standards, as the study mentioned.

Answer: We have addressed your concerns by revising the manuscript to give a more accurate description of the analyses conducted.

In response to your feedback, we have renamed Section 2.4 to "Phytochemical Characterization" and Section 3.1 to "Phytochemical Profile" to better represent the nature of the work described in these sections. We agree that the term "bioactive" should be associated with experimental evidence of biological activity, which is beyond the scope of LC-MS-based chemical characterization.

These changes clarify that the focus of Sections 2.4 and 3.1 is on the chemical characterization and profiling of phytochemical compounds rather than their bioactivity.

We appreciate your input, which has been instrumental in improving the precision and accuracy of our manuscript.

  1. Lines 174-211: There is a difference between "antioxidant capacity" (a thermodynamic assessment of radical scavenging assays) and "antioxidant activity" (a kinetic evaluation), and this study involves both. The manuscript should refer to the correct term when appropriate. Be consistent throughout the manuscript.

Answer: Thank you for your valuable feedback. We have thoroughly revised the manuscript to ensure the correct use of the terms "antioxidant capacity" (reflecting a thermodynamic assessment of radical scavenging) and "antioxidant activity" (indicating a kinetic evaluation). We have maintained consistency in terminology throughout the document to accurately describe the assays used in this study.

  1. Line 349: Was the data normality verified to support a parametric test? Revise and add the relevant information to the manuscript.

Answer: Thank you for your valuable feedback. We have made the necessary revisions to the "Statistical Analysis" section to address your concerns regarding the verification of data normality for parametric tests.

The updated section now includes information on how the normality of the data distributions was assessed before applying parametric tests. Specifically, we conducted the Shapiro-Wilk test to verify data normality, which is now detailed in the revised manuscript as follows:

“Data analysis was performed using R software version 4.2.0 (R Development Core Team, 2011). Before conducting parametric tests, the normality of data distributions was assessed using the Shapiro-Wilk test. Analysis of variance (one-way ANOVA, p < 0.05) was employed to study the matrix effect on antioxidant capacity and activity of samples, provided that the data met the assumptions of normality and homogeneity of variances. Tukey-Kramer comparison tests were then conducted to study the matrix effect on the antioxidant capacity and activity of samples. To analyze correlations between different biological capacities and activities assessed, Pearson’s correlation analysis was performed using R software. The results were visualized as a correlation matrix in the form of a heatmap. A significance level of α = 0.01 was applied to determine statistically significant correlations. All statistical procedures were carried out ensuring that the assumptions for parametric testing were met, with normality verified prior to ANOVA and correlation analyses.”

  1. Table 1: How were the dihydroquercetin isomers quantified in ug/g extract if the correct isomer was not determined? Which standard was used for such quantifications if different per isomer per cultivar? These explanations must be clearly added to the manuscript to avoid confusing interpretations.

Answer: Thank you for your insightful question regarding the quantification of dihydroquercetin isomers in our study. We appreciate the opportunity to clarify this aspect of our methodology.

In our study, we indeed quantified dihydroquercetin isomers in terms of µg/g extract. While we did not determine the exact isomeric identity of each dihydroquercetin peak, we used a standard dihydroquercetin compound for quantification. The quantification was carried out by comparing the peak areas of the dihydroquercetin isomers in the samples to the peak area of the standard as quercetin. This approach is a common practice when isomer-specific standards are not available, and it provides an estimate of the total dihydroquercetin content in the extracts. Although different isomers may have different responses, using a common standard like quercetin allows for a relative comparison of the total content across different samples. We acknowledge that this method does not account for potential differences in the response factors of different isomers, but it is a standard practice in the absence of isomer-specific standards. We based our approach on established methods from the literature where dihydroquercetin is often quantified as a total compound in the absence of specific isomeric standards.

We hope this explanation addresses your concerns regarding the quantification of dihydroquercetin isomers and the use of standards for these measurements. Thank you for helping us improve the clarity and accuracy of our manuscript.

Comments on the Quality of English Language

Some grammar and stylistic issues must be deeply revised. Detailed scrutiny should be performed throughout the manuscript to revise such issues.

Answer: We have undertaken a comprehensive review and revision of the manuscript to address the grammatical and stylistic issues you identified. This involved a detailed scrutiny of the entire text to ensure clarity, coherence, and readability. We believe that these extensive revisions have significantly improved the quality of the manuscript. We are committed to maintaining high standards of scientific communication and appreciate your role in helping us achieve this.

Thank you once again for your valuable input. We hope the revised manuscript meets the journal’s standards and addresses all your concerns.